# Fast Estimation and Optimization of Resistance Diameter on Graphs

## ABSTRACT

The resistance diameter of a graph is the maximum resistance distance among all pairs of nodes in the graph, which has found various applications in many scenarios. However, direct computation of resistance diameter involves the pseudoinverse of graph Laplacian, which takes cubic time and is thus infeasible for huge networks with millions of nodes. In this paper, we consider the computation and optimization problems for resistance diameter of a graph. First, we develop a nearly linear time algorithm to approximate the resistance diameter, which has a theoretically guaranteed error. Then, we propose and study an optimization problem of adding a fixed number of edges to a graph, such that the resistance diameter of the resulting graph is minimized. We show that this optimization problem is NP-hard, and that the objective function is non-supermodular but monotone. Moreover, we propose two fast heuristic algorithms to approximately solve this problem. Finally, we conduct extensive experiments on different networks with sizes up to one million nodes, demonstrating the superiority of our algorithms in terms of efficiency and effectiveness.

## KEYWORDS

Resistance distance, Resistance diameter, Combinatorial optimization, Graph mining, Linear algorithm, Convex hull, Laplacian solver

## 1 INTRODUCTION

Effective resistance, as a fundamental graph metric, has found wide-ranging theoretical and practical applications. In theory, resistance distance has become a cornerstone of algorithmic graph theory, leading to breakthroughs in fast algorithms for problems such as spectral graph sparsification [57], maximum flow approximation [16], random spanning trees generation [41], maximizing the number of spanning trees [36], and solving traveling salesman problems [4]. In practical scenarios, resistance distance has been applied to diverse fields like collaborative recommendation [21], graph embedding [11], image segmentation [7], mining important nodes and edges [10, 37, 55], and link prediction [42]. In data management, it has been used in graph processing systems [52], density-based clustering [56], and localizing anomalous changes [58]. Recently, new methods for evaluating resistance distance have emerged [39, 65], underscoring its significance in both theory and application [19].

Alongside resistance distance, shortest path distance is another key graph metric, defined as the length of the shortest path between two nodes. The diameter of a graph, which is the maximum shortest-path distance, is an important measure with applications in graph analytics [70], privacy preservation [47], CPU cache efficiency [62], community search [40], and community discovery [17]. While shortest path distance focuses on a single path, resistance distance accounts for all paths between two nodes, often leading to more effective applications [10, 23]. For instance, current flow closeness centrality based on resistance distance shows greater

discrimination power than its shortest-path counterpart [37]. A key measure based on resistance distance is resistance diameter, which has numerous applications, including clustering in graphs [3] and hypergraphs [2], distributed clock synchronization over wireless networks [28], and minimizing the Kirchhoff index [30]. However, computing resistance diameter for large graphs is computationally expensive, and efficient algorithms for this task are still lacking. Moreover, reducing resistance diameter, for example by adding edges, is of significant interest in many applications where a smaller resistance diameter is desirable.

In this paper, we study the computation and optimization problems for the resistance diameter $R(\mathcal{G})$. The main contributions and work of this paper are summarized as follows.

- We propose a fast $\epsilon$-approximation algorithm FastRD to evaluate the resistance diameter $R(\mathcal{G})$ of a graph $\mathcal{G}$, which is based on the Johnson-Lindenstrauss Lemma, Laplacian solvers and convex hull. The algorithm has a nearly linear time complexity $\widetilde{O}\left((m + nl)/\epsilon^2\right)$, where the $\tilde{O}(\cdot)$ notation suppresses the poly$(\log n)$ factors, $\epsilon > 0$ is the error parameter, and $l$ is the number of nodes in the boundary of the approximated convex hull.

- We propose and study the following optimization problem: Given a graph $\mathcal{G}$ and an integer $k \ll n$, how to select a set $P$ of $k$ edges in set $Q = (V \times V)\backslash E$ to add to $\mathcal{G}$ such that the resistance diameter of the resulting graph is minimized. We show that this problem is NP-hard, and that the objective function is not supermodular.

- We propose two efficient greedy heuristic algorithms MinDiaEi and MinDiaCH to approximately solve the combinatorial optimization problem. The former has time complexity $\widetilde{O}(km)$, while the latter has complexity $\widetilde{O}\left(k(m + nl)/\epsilon^2\right)$ for a given error parameter $\epsilon > 0$.

- We execute extensive numerical experiments on a large variety of real and model networks to evaluate the performance of our proposed algorithms: FastRD for computing resistance diameter, and MinDiaEi and MinDiaCH for minimizing resistance diameter by edge addition. The obtained experimental results demonstrate that our three approximation algorithms are efficient and accurate. Particularly, they are scalable to large graphs with over one million nodes. Moreover, MinDiaEi and MinDiaCH outperform several baseline strategies for edge selection.

## 2 RELATED WORK

In this section, we briefly review the literature related to our work. The resistance diameter can be seen as an extension of the shortest-path-based diameter, as they are equivalent in tree-like graphs. The diameter of a graph represents the greatest distance between any two nodes, reflecting the maximum time needed for information or matter to propagate. Graph diameter has found applications in

data management and mining, including route recommendation on road networks [66] and reachability querying in large graphs [15]. Significant efforts have been made to design efficient algorithms for computing graph diameter [38, 61], as well as strategies for optimizing it, such as reducing graph diameter by adding edges [1, 18, 22]. However, since resistance diameter accounts for all paths between nodes, methods for computing or optimizing shortest-path diameter do not directly apply to resistance diameter.

Fast computation of resistance distance is essential for many applications, prompting the development of various algorithms, such as random projection methods [43, 57], accelerated by Wilson's algorithm [29]. Local algorithms using random walks and spanning trees [48], as well as Monte Carlo approaches [39, 65], have been proposed to improve performance. While these methods are efficient for computing node-to-node resistance distances, computing resistance diameter, which involves calculating distances for $O(n^2)$ node pairs, remains impractical for large graphs. Although there is no closed-form expression for the resistance diameter of general graphs, explicit formulas have been derived for special structures like hierarchical graphs [50], Hanoi graphs [54], Sierpiński gaskets [31], and Koch networks [69, 73]. Additionally, the scaling behavior of resistance diameter has been studied across various graph topologies, showing linear growth in path and cycle graphs [67], sub-linear growth in hierarchical graphs [54], and logarithmic growth in complete trees and torus graphs [8, 75]. Interestingly, for most real-world networks, resistance diameter and average resistance distance do not depend on the number of nodes [64, 68].

In terms of optimization, prior work has focused on minimizing resistance distance or the Kirchhoff index by edge addition [27]. Researchers have developed algorithms to minimize resistance distance between specific node pairs [13] and to optimize the Kirchhoff index [51, 59]. However, no previous methods directly address minimizing resistance diameter by adding edges, despite edge addition being widely used in other graph editing applications, such as improving node centrality [55] or maximizing overall opinion dynamics [74]. Thus, minimizing resistance diameter through graph edits remains an open and challenging problem.

## 3 PRELIMINARIES

### 3.1 Graph and Related Matrices

Let $\mathcal{G} = (V, E)$ denote a connected undirected unweighted graph with node/vertex set $V$ and edge set $E$. The numbers of nodes and edges in $\mathcal{G}$ are $n = |V|$ and $m = |E|$, respectively. For graph $\mathcal{G} = (V, E)$, let $Q = (V \times V) \backslash E$ be the set of edges that are nonexistent in $\mathcal{G}$. For a nonempty edge set $A \subset Q$, we use $\mathcal{G}(A) = (V, E \cup A)$ to denote the augmented graph of $\mathcal{G} = (V, E)$ with the same node set $V$ as $\mathcal{G}$ but more edges than $\mathcal{G}$.

All the eigenvalues of Laplacian matrix $L$ are non-negative, with zero being the unique eigenvalue. Let $0 = \lambda_1 < \lambda_2 \leq \cdots \leq \lambda_n$ denote its $n$ eigenvalues, and let $u_k, k = 1, 2, \ldots, n$, stand for their corresponding mutually orthogonal unit eigenvectors. The second smallest eigenvalue $\lambda_2$ of matrix $L$ is often called the algebraic connectivity of the graph $\mathcal{G}$ and denoted as $\lambda_2(\mathcal{G})$, and its corresponding normalized eigenvector is called the Fiedler vector of $\mathcal{G}$. Laplacian matrix $L$ can be decomposed as $L = \sum_{k=1}^{n} \lambda_k u_k u_k^\top$.

Note that $L$ is singular, since it has a eigenvalue 0. Thus, $L$ cannot be inverted. Alternatively, we use the Moore-Penrose generalized inverse of $L$, which is simply called pseudo-inverse of $L$ [9]. Let $L^\dagger$ denote the pseudo-inverse of $L$, which can be expressed as $L^\dagger = \sum_{k=2}^{n} \frac{1}{\lambda_k} u_k u_k^\top$.

The symmetry of both matrices $L$ and its pseudo-inverse $L^\dagger$ implies that $L$ and $L^\dagger$ share identical null space [9], obeying $L\mathbf{1} = \mathbf{0}$ and $L^\dagger \mathbf{1} = \mathbf{0}$. Consider the relation $\mathcal{J} = \mathbf{1}\mathbf{1}^\top$, we obtain $L\mathcal{J} = \mathcal{J}L = L^\dagger \mathcal{J} = \mathcal{J}L^\dagger = \mathbf{O}$. Applying the the spectral decompositions for matrices $L$ and $L^\dagger$, the pseudoinverse $L^\dagger$ of $L$ can be obtained to be [27]

$$L^\dagger = \left(L + \frac{1}{n}\mathcal{J}\right)^{-1} - \frac{1}{n}\mathcal{J}. \tag{1}$$

### 3.2 Electrical Networks, Resistance Distance and Resistance Diameter

By replacing every edge in graph $\mathcal{G}$ with a unit resistance, we obtain an electrical network [20] associated with graph $\mathcal{G}$. In the case incurring no confusion, for a graph $\mathcal{G}$, we also use $\mathcal{G}$ to denote its corresponding electrical network. The resistance distance $r(u, v)$ between two nodes $u$ and $v$ in graph $\mathcal{G} = (V, E)$ is defined as the effective resistance between $u$ and $v$ in the corresponding electrical network [33], which can be expressed in terms of the entries of matrix $L^\dagger$ as

$$r(u, v) = L_{uu}^\dagger + L_{vv}^\dagger - 2L_{uv}^\dagger. \tag{2}$$

For a graph $\mathcal{G} = (V, E)$, the maximum value of resistance distances over all its node pairs is called the resistance diameter and is denoted by $R(\mathcal{G})$. That is,

$$R(\mathcal{G}) = \max_{u,v \in V} r(u, v). \tag{3}$$

## 4 ALGORITHMS FOR COMPUTING RESISTANCE DIAMETER

In this section, we introduce three algorithms for computing resistance diameter.

### 4.1 Exact Computation by Matrix Inverse

By using (1), (2), and (3), we can compute the exact resistance diameter for any graph $\mathcal{G}$. First, using (1) we obtain the pseudoinverse $L^\dagger$ of matrix $L$ by inverting the matrix $L + \frac{1}{n}\mathcal{J}$ in $O(n^3)$ time. Then, according to (2), we compute the resistance distance $r(u, v)$ for all the $O(n^2)$ pairs of nodes $u$ and $v$ in $\mathcal{G}$, which takes time $O(n^2)$. Finally, we use (3) to find the resistance diameter $R(\mathcal{G})$ of graph $\mathcal{G}$, which runs in $O(n^2)$ time. The rigorous algorithm is called EXACT, which is described in Algorithm 1. It is obvious that the total running time of EXACT is $O(n^3)$.

### 4.2 Algorithm Based on JL Lemma and Laplacian Solver

As shown above, the most time-consuming step in the Algorithm 1 for computing resistance diameter is inverting the matrix $L + \frac{1}{n}\mathcal{J}$, which requires $O(n^3)$ time. To avoid this time-consuming matrix inverse operation, here we describe a randomized algorithm, which approximates the resistance diameter in $O(n^2)$ time.

 

---

**Algorithm 1:** Exact($\mathcal{G}$)

**Input** : A connected graph $\mathcal{G} = (V, E)$ with Laplacian matrix $L$

**Output** : $R(\mathcal{G})$ : The resistance diameter of graph $\mathcal{G}$

1 Compute the pseudoinverse $L^\dagger$ of $L$ by

$$L^\dagger = \left(L + \frac{1}{n}\mathcal{J}\right)^{-1} - \frac{1}{n}\mathcal{J}$$

2 Compute $r(u, v)$ for all node pairs in $\mathcal{G}$ by

$$r(u, v) = L_{uu}^\dagger + L_{vv}^\dagger - 2L_{uv}^\dagger$$

3 $R(\mathcal{G}) = \max_{u,v \in V} r(u, v)$

4 **return** $R(\mathcal{G})$

---

Before introducing our algorithm, we first recast the resistance distance $r(u, v)$ as [57]

$$r(u, v) = \left\| BL^\dagger (e_u - e_v) \right\|_2^2. \tag{4}$$

In other words, we can embed the $n$ nodes in graph $\mathcal{G}$ to $n$ points corresponding to $n$ $m$-dimension vectors $BL^\dagger e_i$, $i = 1, 2, \ldots, n$, in Euclidean space $\mathbb{R}^m$, which preserve the resistance distances. For two vectors $BL^\dagger e_u$ and $BL^\dagger e_v$ in $\mathbb{R}^m$, let $d(u, v)$ denote their Euclidean distance. Then, we have $r(u, v) = d(u, v)^2$.

Equation (4) reduces the computation of resistance distance $r(u, v)$ to the calculation of the $\ell_2$ norms $\left\| BL^\dagger (e_u - e_v) \right\|_2^2$ of two vectors in $\mathbb{R}^m$. However, the complexity for exactly computing this $\ell_2$ norms is still high, since the dimension $m$ of vectors $BL^\dagger e_i$ ($i = 1, 2, \ldots, n$) is high and it still needs inverting matrix $L + \frac{1}{n}\mathcal{J}$ to obtain $L^\dagger$.

We first alleviate the computation burden by reducing the dimension of vectors. For this purpose, we apply the Johnson-Lindenstrauss (JL) Lemma [32] to approximate the $\ell_2$ norms. For $\left\| BL^\dagger (e_u - e_v) \right\|_2^2$, if we project the set of $n$ $m$-dimension vectors, i.e., the $n$ column vectors of matrix $BL^\dagger$, onto a low $d$-dimension subspace spanned by the columns of a random matrix $Q \in \mathbb{R}^{d \times m}$ with entries being $\pm 1/\sqrt{d}$, where $d = \lceil 24 \log(n)/\epsilon^2 \rceil$ for given $\epsilon$, then we get an $\epsilon$-approximation of $\left\| BL^\dagger (e_u - e_v) \right\|_2^2$ with high probability. For consistency, we introduce the JL Lemma [32].

LEMMA 4.1. *(JL Lemma) Given fixed vectors $v_1, v_2, \ldots, v_n \in \mathbb{R}^m$ and $\epsilon > 0$, let $Q_{d \times m}$, $d \geq 24 \log n/\epsilon^2$, be a matrix, each entry of which is equal to $1/\sqrt{d}$ or $-1/\sqrt{d}$ with the same probability $1/2$. Then with probability at least $1 - 1/n$,*

$$(1 - \epsilon)\left\| v_i - v_j \right\|_2^2 \leq \left\| Qv_i - Qv_j \right\|_2^2 \leq (1 + \epsilon)\left\| v_i - v_j \right\|_2^2$$

*for all pairs $i, j \leq n$.*

Let $Q_{d \times m}$ be a random $\pm 1/\sqrt{d}$ matrix where $d = \lceil 24 \log(n)/\epsilon^2 \rceil$. By Lemma 4.1, for any $u, v \in V$ we have

$$r(u, v) \overset{\epsilon}{\approx} \left\| QBL^\dagger (e_u - e_v) \right\|_2^2. \tag{5}$$

Thus, if we embed the $n$ vertices in graph $\mathcal{G}$ to $n$ vectors $QBL^\dagger e_i$, $i = 1, 2, \ldots, n$, in a low-dimension space $\mathbb{R}^d$, the resistance distance for any node pair is approximately preserved.

However, the computation of the $n$ vectors $QBL^\dagger e_i$ ($i = 1, 2, \ldots, n$) still requires the pseudoinverse $L^\dagger$, which involves inverting matrix

---

**Algorithm 2:** ApproxRD($\mathcal{G}, \epsilon$)

**Input** : A connected graph $\mathcal{G} = (V, E)$, a parameter $\epsilon$

**Output** : $\bar{R}(\mathcal{G})$ : The approximate resistance diameter of graph $\mathcal{G}$

1 $d = \lceil 24 \log n/\epsilon^2 \rceil$

2 $\tilde{X}_{d \times n} \leftarrow$ ApproxER($\mathcal{G}, \epsilon$)

3 Compute $\tilde{r}(u, v)$ for all node pairs in $\mathcal{G}$ by

$$\tilde{r}(u, v) = ||\tilde{X}(e_u - e_v)||_2^2$$

4 $\bar{R}(\mathcal{G}) \leftarrow \max_{u,v \in V} ||\tilde{X}(e_u - e_v)||_2^2$

5 **return** $\bar{R}(\mathcal{G})$

---

$L + \frac{1}{n}\mathcal{J}$. In order to avoid matrix inversion, one can resort to the fast symmetric, diagonally-dominant (SDD) linear system solver [35] to evaluate $QBL^\dagger e_i$, since $L$ is an SDDM matrix. In the sequel, for the convenience of description, we use the notation $\tilde{O}(\cdot)$ to hide poly log factors.

LEMMA 4.2. *There is an algorithm $x = LaplSolve(L, z, \delta)$ which takes a Laplacian matrix $L$, a column vector $z$, and an error parameter $\delta > 0$, and returns a column vector $x$ satisfying $1^\top x = 0$ and*

$$\left\| x - L^\dagger z \right\|_L \leq \delta \left\| L^\dagger z \right\|_L.$$

*The algorithm runs in expected time $\tilde{O}(m \log(1/\delta))$.*

Based on the Laplacian solvers and the JL Lemma, an approximation algorithm ApproxER was proposed in [57] to estimated resistance distance in nearly linear time with respect to the number of edges, as stated in Lemma 4.3.

LEMMA 4.3. *There is an $\tilde{O}\left(m \log(1/\delta)/\epsilon^2\right)$-time algorithm which on input $\epsilon > 0$, $\delta \leq \frac{\epsilon}{3}\sqrt{\frac{2(1-\epsilon)}{(1+\epsilon)n^3}}$ and $\mathcal{G} = (V, E)$ computes a $\lceil 24 \log n/\epsilon^2 \rceil \times n$ matrix $\tilde{X}$ such that with probability at least $1 - 1/n$,*

$$r(u, v) \overset{\epsilon}{\approx} ||\tilde{X}(e_u - e_v)||_2^2$$

*for every pair of vertices $u, v \in V$.*

By using Lemma 4.3, we develop an approximation algorithm ApproxRD estimating the resistance diameter $R(\mathcal{G})$ for graph $\mathcal{G}$ in $O(n^2)$ time. The pseudocode of ApproxRD is provided in Algorithm 2.

### 4.3 Fast Algorithm Based on Convex Hull

The above approximation algorithm ApproxER based on the Laplacian solvers and the JL Lemma is still infeasible for large networks, since it runs in $O(n^2)$ time. This is caused by the computation of the resistance distances for all the $O(n^2)$ pairs of nodes in $\mathcal{G}$. In this subsection, we will show that one only need to compute the resistance distances for a small number of node pairs, since we are only concerned with the resistance diameter $R(\mathcal{G})$. We then develop a fast algorithm, which approximates $R(\mathcal{G})$ of an arbitrary graph $\mathcal{G}$. Our algorithm has a nearly linear time and space cost with respect to the number of edges. Furthermore, it has a guaranteed theoretical error with high probability.

Equation (5) reduces the approximation estimation of resistance distances to the calculation of the distances between the $O(n^2)$ pairs of $n$ points $QBL^\dagger e_i$ ($i = 1, 2, \ldots, n$) in Euclidean space $\mathbb{R}^d$. To

calculate the resistance diameter $R(\mathcal{G})$, we need to find the pair of nodes with the largest resistance distance. Equation (5) shows that we only to find the distance of the pair of farthest points in $d$-dimensional Euclidean space, which can be solved by the convex hull technique.

**DEFINITION 4.4.** *[49] Given a set of $n$ points $S = \{v_1, v_2, \ldots, v_n\}$ in $\mathbb{R}^d$, its convex hull is the (unique) minimal convex polytope containing $S$.*

For a given convex hull of point set $S$, its boundary is denoted by $C(S)$, and the subset of points in $S$, which lie on the boundary of the convex hull is denoted by $\bar{S}$. By Definition 4.4, it can be observed that in an Euclidean space the pair of nodes with the maximum distance in a point set must lie on the boundary of the convex hull corresponding to that point set.

There are many methods to find the convex hull of a point set of $n$ points in a low-dimensional Euclidean space. However, the determination of the samples in the convex hull of a point set of high dimensions is a time-complex task. For dimensions $d > 3$, the time for computing the convex hull is $O(n^{\lfloor d/2 \rfloor})$, matching the worst-case output complexity of the problem [14]. Therefore, computing the convex hull of $n$ points in $d$-dimension space is time-consuming. Fortunately, there is a fast algorithm APPROXCH approximating the convex hull [5]. Before introducing the algorithm APPROXCH, we provide some more definitions. For any given set $S = \{v_i \in \mathbb{R}^d : i = 1, 2, \ldots, n\}$, let $D(S)$ denote the diameter of $S$. That is, $D(S) = \max_{v_i, v_j \in S} ||v_i - v_j||_2$, which is the maximum distance between all pairs of points $v_i$ and $v_j$, obeying relation $D(S) = D(\bar{S})$.

**LEMMA 4.5.** *[5] There is an algorithm $\hat{S} = APPROXCH(S, \theta)$ which takes a set $S = \{v_i \in \mathbb{R}^d : i = 1, 2, \ldots, n\}$ and an error parameter $\theta \in (0, 1)$, and returns an $l$-node subset $\hat{S}$ of $\bar{S}$. The algorithm runs in $O(nl(d + \theta^{-2}))$ time, and the Euclidean distance for any $p \in \bar{S}$ to $C(\hat{S})$ is at most $\theta D(S)$, where $C(\hat{S})$ is the boundary of convex hull of $\hat{S}$.*

Lemma 4.5 shows that using the APPROXCH algorithm, we can obtain an approximate point set $\hat{S}$ for $\bar{S}$, the boundary of convex hull $C(S)$ in a $d$-dimensional Euclidean space. Moreover, APPROXCH provides an upper bound for the distance between any point in $\bar{S}$ and any point in $C(\hat{S})$. Based on Lemma 4.5, we can show that $D(\hat{S})$ is a good approximation for $D(\bar{S})$, as stated in Lemma 4.6.

**LEMMA 4.6.** *Given a point set $S = \{v_i \in \mathbb{R}^d : i = 1, 2, \ldots, n\}$, a parameter $\theta = \frac{\epsilon}{12}$, a subset $\bar{S} \subseteq S$, whose points lie on the boundary of the convex hull $C(S)$ of set $S$, and $\hat{S} = APPROXCH(S, \theta)$ that is a $l$-node subset of $\bar{S}$. Let $x, y \in \hat{S}$, $u, v \in \bar{S}$, $d(x, y) = D(\hat{S})$, and $d(u, v) = D(\bar{S})$. Then, we have*

$$d(x, y) \overset{\frac{\epsilon}{6}}{\approx} d(u, v). \tag{6}$$

Making use of Eq. (4), Lemma 4.3, and Lemma 4.6, we obtain the following result.

**LEMMA 4.7.** *Given a graph $\mathcal{G} = (V, E)$, a set $S = \{v_i \in \mathbb{R}^m : i = 1, 2, \ldots, n\}$, where $v_i = \mathbf{QBL}^\dagger e_i$, a parameter $\epsilon > 0$, and $\theta = \frac{\epsilon}{12}$. Suppose that $\hat{S} = APPROXCH(S, \theta)$ is an $l$-point subset of $\bar{S}$ and*

---

**Algorithm 3:** FASTRD$(\mathcal{G}, \epsilon)$

   **Input** : A connected graph $\mathcal{G} = (V, E)$, a parameter $\epsilon$
   **Output** : $\hat{R}(\mathcal{G})$: Approximation of resistance diameter
           $R(\mathcal{G})$ of graph $\mathcal{G}$

1   $d = \lceil 24 \log n / \epsilon^2 \rceil$, $\theta = \frac{\epsilon}{12}$
2   $\tilde{X}_{d \times n} \leftarrow$ APPROXER$(\mathcal{G}, \epsilon)$
3   $S \leftarrow \{s_i \in \mathbb{R}^d | s_i = \tilde{X}_{[:, i]}, i = 1, 2, \ldots, n\}$
4   $\hat{S} \leftarrow$ APPROXCH$(S, \theta)$
5   Compute $\tilde{r}(u, v)$ for all node pairs in $\hat{S}$ by
    $\tilde{r}(u, v) = ||\tilde{X}(\boldsymbol{e}_u - \boldsymbol{e}_v)||_2^2$
6   $\hat{R}(\mathcal{G}) \leftarrow \max_{u, v \in \hat{S}} \tilde{r}(u, v)$
7   **return** $\hat{R}(\mathcal{G})$

---

$\bar{R}(\mathcal{G}) = APPROXRD(\mathcal{G}, \epsilon)$. Let $\hat{R}(\mathcal{G}) = D(\hat{S})^2 = \max_{u, v \in \hat{S}} d(u, v)^2$. Then, we have

$$\hat{R}(\mathcal{G}) \overset{\epsilon/3}{\approx} \bar{R}(\mathcal{G}). \tag{7}$$

Since $\hat{S}$ contains $l$ points, we can obtain $\hat{R}(\mathcal{G})$ by computing the distances between $l^2$ pairs of points, instead of $O(n^2)$ pairs of points. In general, $l$ is much smaller than $n$, using Lemma 4.7 to evaluate the resistance diameter of a graph significantly reduces the computation time.

Based on the above results, we are now in position to propose a fast algorithm FASTRD approximating the resistance diameter of $\mathcal{G}$, the pseudocode of which is presented in Algorithm 3. In FASTRD, $\tilde{X}$ is a $\lceil 24 \log n / \epsilon^2 \rceil \times n$ matrix, $S$ is a set of points, which are the column vectors of matrix $\tilde{X}$. Set $\hat{S}$ is an approximation of $\bar{S}$, which is the set of points on the boundary of the convex hull $C(S)$. The performance of the algorithm FASTRD is given in Theorem 4.8.

**THEOREM 4.8.** *The algorithm FASTRD$(\mathcal{G}, \epsilon)$ runs in $\widetilde{O}\left((m + nl)/\epsilon^2\right)$ time, and outputs an approximation value $\hat{R}(\mathcal{G})$ of resistance diameter $R(\mathcal{G})$ for graph $\mathcal{G}$, satisfying*

$$(1 - \epsilon)R(\mathcal{G}) \leq \hat{R}(\mathcal{G}) \leq (1 + \epsilon)R(\mathcal{G}).$$

## 5 MINIMIZING RESISTANCE DIAMETER BY EDGE ADDITION

In this section, we formulate and study the problem of adding a fixed number edges in order to minimize the resistance diameter of a graph.

### 5.1 Problem Statement, Optimal Solution, and Simple Greedy Algorithm

It is well-known that [13], adding an edge to a graph does not increase the resistance distance between any pair of nodes, including the resistance diameter. This motivates us to study the problem of how to choose a fixed number $k$ of nonexistent edges to a graph in order to minimize the resistance diameter of the resultant new graph. The problem is mathematically formulated as follows.

**PROBLEM 1.** *(Resistance Diameter Minimization) Given a graph $\mathcal{G} = (V, E)$ with $n$ nodes and $m$ edges and a candidate edge set $Q = (V \times V) \backslash E$, for any integer $1 \leq k \leq |Q|$, find an edge subset*

---

**Algorithm 4:** SIMPLE($\mathcal{G}, Q, k$)

**Input** : A connected graph $\mathcal{G} = (V, E)$, a candidate edge set $Q$, an integer $1 \leq k \leq |Q|$

**Output** : $P$ : A subset of $Q$ with $|P| = k$ edges

1 Initialize solution $P = \emptyset$

2 **for** $i = 1$ *to* $k$ **do**

3     Select $e_i$ s.t. $e_i \leftarrow \arg\min_{e \in Q \backslash P}$ EXACT$(\mathcal{G}(\{e\}))$

4     Update solution $P \leftarrow P \cup \{e_i\}$

5     Update the graph $\mathcal{G} \leftarrow \mathcal{G}(\{e_i\})$

6 **return** $P$

---

$P^* \subseteq Q$ such that the resistance diameter of $\mathcal{G}(P^*) = (V, E \cup P^*)$ is minimized, that is,

$$P^* \in \arg\min_{P \subseteq Q, |P| = k} R(\mathcal{G}(P)).$$

Problem 1 is a intrinsically combinatorial one subject to a cardinality constraint. We can obtain its optimal solutions by exhausting all the $\binom{|Q|}{k}$ subsets $P$ with $k$ edges. For each subset, we calculate the resistance diameter of the resultant graph, which requires $O(n^3)$ time. Then, output the optimal solution, which minimizes the resistance diameter. Although this method is simple, it is computationally impossible even for small networks, since its has an exponential complexity $O(\binom{|Q|}{k} n^3)$.

To tackle the exponential complexity of brute-force search, one often resorts to greedy heuristics. Below we present a simple greedy algorithm for Problem 1, which is outlined in Algorithm 4 and described as follows. Initially, we set the set $P$ of added edges to be empty, then $k$ edges are added from set $Q \backslash P$ iteratively. In each iteration step $i$, edge $e_i$ in set $Q \backslash P$ of candidate edges is selected, which minimizes the resistance diameter of the new graph. The algorithm terminates when $k$ edges are selected to be added to $P$. For every candidate edge, it needs computing resistance diameter of a resultant graph. A direct calculation of resistance diameter requires $O(n^3)$ time, leading to a total computation complexity $O(k|Q|n^3)$.

We also address the hardness of the resistance diameter minimization problem and the non-supermodular property of the objective function, with additional details provided in the appendix.

## 5.2 Two Fast Approximation Algorithms

Although the simple greedy algorithm in Algorithm 4 is much faster than the brute-force algorithm, it is not applicable to large-scale networks since it takes too much time to calculate $R(\mathcal{G}(\{e\}))$ for each edge $e \in Q \backslash P$ in cubic time $O(n^3)$ in each iteration. Then, it take quadratic time $O(n^2)$ to obtain the edge $Q \backslash P$ that minimizes the resistance diameter of resultant graph. To reduce the computational complexity, we propose two nearly linear time approximation algorithms. The former iteratively find $k$ edges to minimize an upper bound of the resistance diameter, while the latter finds the furthest $k$ pairs of nodes and add edges among them.

*5.2.1 Fast Alternative Algorithm Minimizing an Upper Bound of Resistance Diameter.* We first approximately solves Problem 1 from the perspective of spectral graph theory. To this end, we propose a

---

**Algorithm 5:** MINDIAEI($\mathcal{G}, Q, k, \delta$)

**Input** : A connected graph $\mathcal{G} = (V, E)$, a candidate edge set $Q$, an integer $1 \leq k \leq |Q|$, a parameter $\delta$

**Output** : $P$ : A subset of $Q$ with $|P| = k$ edges

1 Initialize solution $P = \emptyset$

2 **for** $i = 1$ *to* $k$ **do**

3     Let $L$ be the Laplacian matrix of graph $\mathcal{G}$

4     Compute $(\lambda, \boldsymbol{u}) = $ EIGENPAIR$(L, \delta)$

5     Select $e_i$ s.t. $e_i \leftarrow \arg\max_{(j,t) \in Q \backslash P}(\boldsymbol{u}_j - \boldsymbol{u}_t)^2$

6     Update solution $P \leftarrow P \cup \{e_i\}$

7     Update the graph $\mathcal{G} \leftarrow \mathcal{G}(\{e_i\})$

8 **return** $P$

---

fast approximation algorithm to minimize an upper bound of the resistance diameter, as an alternative way to address Problem 1. Note that the approach of optimizing bounds of a quantity, rather than the quantity itself, has been previously explored in the literature [44, 60]. Since $2/\lambda_2$ is an upper bound for the resistance diameter $R(\mathcal{G})$[63], Problem 1 can be reduced to maximizing $\lambda_2$ by adding $k$ nonexistent edges to graph $\mathcal{G}$.

Note that the problem of maximizing $\lambda_2$ by adding $k$ edges is combinatorial and can be solved exactly by exhaustive search. It is easy to see that there are $\binom{|Q|}{k}$ choices for edge selection. For each of $\binom{|Q|}{k}$ cases, we compute $\lambda_2$ of the corresponding Laplacian matrix and return the set of $k$ edges maximizing $\lambda_2$. However, this is not practical for large $|Q|$ and $k$. Therefore, we use the efficient heuristic algorithm proposed in [26]. Concretely, this heuristic method adds the $k$ edges one at a time. At each time, it chooses the edge $(i, j)$ which has the largest value of $(\boldsymbol{u}_i - \boldsymbol{u}_j)^2$, where $\boldsymbol{u}$ is a Fiedler vector of the current graph. Since directly computing the Fiedler vector requires $O(n^3)$ time, we need to explore an efficient method for computing Fiedler vector quickly. Fortunately, this can be solved by the method in [6], as given in the following lemma.

LEMMA 5.1. *[6] For a connected graph $\mathcal{G}$ with the Laplacian matrix $L$, there exists a nearly-linear time $\widetilde{O}(m)$ algorithm $(\lambda, \boldsymbol{u}) = $ EIGENPAIR$(L, \delta)$, whose outputs $\lambda$ and $\boldsymbol{u}$ are, respectively, approximations for algebra connectivity and Fiedler vector for graph $\mathcal{G}$, satisfying $\lambda = \boldsymbol{u}^\top L \boldsymbol{u} \leq (1 + \delta)\lambda_2$.*

Based on Lemma 5.1, we propose a fast algorithm called MINDIAEI to approximately solve Problem 1. The outline of MINDIAEI is presented in Algorithm 5, which has a total running time $\widetilde{O}(km)$.

*5.2.2 Convex Hull Based Fast Iterative Algorithm .* By definition, the resistance diameter of a graph is the largest value of resistance distances between all pairs of nodes in the graph. This enlightens us to minimize the resistance diameter by iteratively adding edges between the pair of nodes with the maximum resistance distance. Specifically, in each iteration step $i$, we only need to find the pair of nodes with the maximum resistance distance in the graph and establish a link between them. Actually, as shown in (4), the resistance distance is equivalent to Euclidean distance between two corresponding points in an Euclidean space. Then, Problem 1 can be approximately solved by finding the pair of points with the maximum distance in an Euclidean space at each iteration step. Based

---

**Algorithm 6:** MinDiaCH($\mathcal{G}, Q, k, \epsilon$)

**Input** : A connected graph $\mathcal{G} = (V, E)$, a candidate edge set $Q$, an integer $1 \leq k \leq |Q|$, a parameter $\epsilon$

**Output** : $P$ : A subset of $Q$ with $|P| = k$ edges

1  Initialize solution $P = \emptyset$

2  $d = \lceil 24 \log n / \epsilon^2 \rceil$, $\theta = \frac{\epsilon}{12}$

3  **for** $i = 1$ *to* $k$ **do**

4     $\tilde{X}_{d \times n} \leftarrow$ ApproxER($\mathcal{G}, \epsilon$)

5     $S \leftarrow \{s_j \in \mathbb{R}^d | s_j = \tilde{X}_{[:,j]}, j = 1, 2, \ldots, n\}$

6     $\hat{S} \leftarrow$ ApproxCH($S, \theta$)

7     Select $e_i$ s.t. $e_i \leftarrow \arg\max_{u,v \in \hat{S}, (u,v) \in Q \setminus P} \|\tilde{X}(\boldsymbol{e}_u - \boldsymbol{e}_v)\|_2^2$

8     Update solution $P \leftarrow P \cup \{e_i\}$

9     Update the graph $\mathcal{G} \leftarrow \mathcal{G}(\{e_i\})$

10  **return** $P$

---

on this idea, we propose a fast algorithm MinDiaCH for Problem 1, the pseudocode of which is illustrated in Algorithm 6.

The algorithm MinDiaCH is based on the ApproxCH algorithm in Lemma 4.5. For $n$ points in a Euclidean space, the points that lie on the boundary of the convex hull of this point set are the farthest from one another. Using ApproxCH, we can obtain a set $\hat{S}$ of $l$ nodes, all of which lie on the boundary of the convex hull. Our algorithm, MinDiaCH, selects the pair of nodes of $\hat{S}$ that are farthest apart at each step, which allows us to identify the pair of nodes in the original graph with the greatest resistance distance. With this approach, we achieve a fast and efficient solution to Problem 1. The total running time of MinDiaCH is $\widetilde{O}\left(k(m + nl)/\epsilon^2\right)$.

## 6 EXPERIMENTS

In this section, we present experimental results to evaluate the performance of the proposed FastRD algorithm for computing the resistance diameter, and MinDiaCH and MinDiaEi for minimizing it by adding a specified number of edges. The source code is publicly available on https://anonymous.4open.science/r/ERdiam-612E.

### 6.1 Experimental Setup

**Datasets and Equipment.** To evaluate the performance of our proposed approximation algorithms, we perform experiments on different realistic networks representatively selected from various domains, which are from Koblenz Network Collection [34] and Network Repository [53]. Table 1 reports the related information for the considered real networks. All our experiments are conducted on a Linux box with an Intel i7-7700K @ 4.2-GHz (4 Cores) and with 128-GB RAM. All algorithms are implemented in *Julia v1.0.3*, where the Laplace Solver is from [35].

### 6.2 Performance of Algorithm FastRD Computing for Resistance Diameter

*6.2.1 Results on Realistic Networks.* We now evaluate the performance of the FastRD algorithm in terms of efficiency and accuracy on real-world networks. For this purpose, we compare our algorithm FastRD with two methods: Exact and GEER [65] that is

the state-of-the-art algorithm for querying resistance distances between node pairs. In [65], the error threshold for the algorithm GEER is set between 0.01 and 0.5. In our experiments, the error threshold of algorithm GEER is set to be 0.1.

We first evaluate the efficiency of the algorithm FastRD. In Table 1, we report the running times of Exact, GEER, and FastRD on several real-world networks. To objectively assess the execution time for the three algorithms on all considered networks, we enforced the program to run on a single thread. From Table 1, it is evident that both Exact and FastRD are faster than GEER. The main reasons are as follows: although GEER is adept at quickly querying the resistance distance between a single node pair, it requires running GEER $O(n^2)$ times to obtain the resistance diameter. Therefore, GEER is only suitable for small networks with fewer than 30,000 nodes. For larger networks, it cannot return results within 24 hours.

From Table 1, we can also observe that for small networks with fewer than 10,000 nodes, Exact is more efficient than FastRD. However, for larger networks and various approximation parameters $\epsilon$, the computational time for FastRD is significantly smaller than that for Exact. For the last seven networks in Table 1, with node counts ranging from $10^6$ to $10^7$, we cannot run the Exact algorithm on our system due to memory and computational constraints. The Exact algorithm requires directly inverting the Laplacian matrix of the graph, which is computationally infeasible for large networks. In contrast, for these networks, we can approximately compute their resistance diameter by applying the FastRD algorithm, further demonstrating that FastRD is efficient and scalable to very large networks.

Except for the high efficiency, algorithm FastRD also provides a good approximation $\hat{R}(\mathcal{G})$ for the resistance diameter $R(\mathcal{G})$. To show the accuracy of FastRD, we compare the approximate values $\hat{R}(\mathcal{G})$ returned by FastRD with the rigorous results of $R(\mathcal{G})$ returned by Exact. In Table 1, we report the relative errors $\sigma$ of algorithm FastRD, where $\sigma$ is defined as $\sigma = (|R(\mathcal{G}) - \hat{R}(\mathcal{G})|/R(\mathcal{G}))$. From Table 1 we can see that the actual relative errors for all $\epsilon$ and all networks are very small, and are almost negligible for smaller $\epsilon$. More interestingly, for all networks tested, $\sigma$ are magnitudes smaller than the theoretical guarantee. Therefore, the approximation algorithm FastRD provides a very accurate result for resistance diameter in practice.

*6.2.2 Results on Model Networks.* For a general graph $\mathcal{G}$, to obtain an exact expression for the resistance diameter $R(\mathcal{G})$ is very difficult and even impossible. However, we can obtain closed-form solution to this quantity for some deterministic model networks, such as the hierarchical graphs [50], the Hanoi graphs [54], the Sierpiński gaskets [31], as well as and the Koch networks [73]. The detailed descriptions of the four model networks, along with their corresponding expressions for resistance diameter, are provided in the appendix.

To further demonstrate the performance of our algorithm FastRD approximating resistance diameter, we apply it to estimate the resistance diameter for the aforementioned four deterministic model networks: hierarchical graphs, Hanoi graphs, Sierpiński Gasket graphs and Koch networks. In Table 2, we report the exact resistance diameter $R(\mathcal{G})$, the approximation $\hat{R}(\mathcal{G})$, relative error

**Table 1: The running time (seconds, $s$) of Exact, GEER and FastRD with various $\epsilon$, as well the relative error $\sigma$ ($\times 10^{-2}$) on various networks. For each network $\mathcal{G}$, we indicate the number of nodes $n$, the number of edges $m$ and the resistance diameter $R(\mathcal{G})$.**

| Network | $n$ | $m$ | $R(\mathcal{G})$ | Running time ($s$) | | FastRD | | Relative error $\sigma$ | FastRD | |
|---|---|---|---|---|---|---|---|---|---|---|
| | | | | Exact | GEER | 0.2 | 0.1 | GEER | 0.2 | 0.1 |
| EmailUN | 1,133 | 5,451 | 4.278 | 0.418 | 129.311 | 4.273 | 7.215 | 0.02 | 0.82 | 0.18 |
| USGrid | 4,941 | 6,594 | 18.283 | 9.357 | 1412.265 | 10.014 | 16.528 | 0.03 | 0.17 | 0.02 |
| Government | 7,057 | 89,429 | 6.207 | 20.331 | 5433.781 | 13.792 | 31.445 | 0.08 | 0.66 | 0.16 |
| Hep-th | 8,361 | 15,751 | 12.154 | 13.112 | 8834.422 | 30.776 | 94.534 | 0.11 | 0.26 | 0.14 |
| Athletes | 13,866 | 86,811 | 6.872 | 89.722 | 31521.387 | 32.452 | 66.587 | 0.55 | 0.87 | 0.56 |
| Musae-facebook | 22,470 | 170,823 | 8.956 | 261.931 | 74842.871 | 113.449 | 150.819 | 0.07 | 0.88 | 0.02 |
| New-sites | 27,917 | 205,964 | 10.016 | 426.714 | – | 96.414 | 159.973 | – | 0.34 | 0.07 |
| Musae-git | 37,700 | 289,003 | 7.522 | 890.925 | – | 182.239 | 220.131 | – | 0.91 | 0.26 |
| HU | 47,538 | 222,887 | 8.018 | 2024.661 | – | 219.461 | 370.527 | – | 0.49 | 0.23 |
| HR | 54,573 | 498,202 | 6.446 | 2334.059 | – | 209.618 | 431.59 | – | 0.74 | 0.55 |
| Epinions | 75,877 | 508,836 | 10.868 | 5555.885 | – | 318.196 | 416.280 | – | 1.05 | 0.87 |
| Slashdot | 77,360 | 828,161 | 5.009 | 5854.217 | – | 386.418 | 574.755 | – | 0.77 | 0.61 |
| Delicious* | 536,108 | 1,365,961 | 3.018 | – | – | 1095.641 | 2062.206 | – | – | – |
| FourSquare* | 639,014 | 3,214,986 | 3.018 | – | – | 1236.412 | 2667.382 | – | – | – |
| Lastfm-song* | 1,085,612 | 19,150,868 | 4.005 | – | – | 3937.847 | 5887.932 | – | – | – |
| Edit* | 1,347,094 | 5,276,371 | 6.31 | – | – | 2343.134 | 3811.039 | – | – | – |
| Wikipedia-growth* | 1,870,709 | 39,953,145 | 6.77 | – | – | 5441.601 | 6891.479 | – | – | – |
| Flixster* | 2,523,386 | 7,918,801 | 8.31 | – | – | 3617.261 | 8587.502 | – | – | – |

**Table 2: Exact resistance diameters $R(\mathcal{G})$, their approximations $\hat{R}(\mathcal{G})$, relative error $\rho = (|R(\mathcal{G}) - \hat{R}(\mathcal{G})|/R(\mathcal{G}))$, and running time (seconds, $s$) for $\hat{R}(\mathcal{G})$ on four different networks.**

| Network | Vertices | Edges | $R(\mathcal{G})$ | $\hat{R}(\mathcal{G})$ | $\rho$ | Time |
|---|---|---|---|---|---|---|
| $\mathcal{H}(13)$ | 1,594,323 | 2,391,483 | 16.667 | 16.498 | 0.0101 | 1472 |
| $\mathcal{T}(12)$ | 1,594,323 | 2,391,483 | 764.656 | 766.947 | 0.0029 | 1521 |
| $\mathcal{S}(13)$ | 2,391,486 | 4,782,969 | 510.437 | 514.247 | 0.0075 | 2303 |
| $\mathcal{K}(10)$ | 2,097,153 | 3,145,728 | 14 | 13.89 | 0.0078 | 2256 |

$\rho = (|R(\mathcal{G}) - \hat{R}(\mathcal{G})|/R(\mathcal{G}))$, and running time (seconds, $s$) for $\hat{R}(\mathcal{G})$ on four deterministic graphs. The corresponding approximation $\hat{R}(\mathcal{G})$ is obtained via our algorithm FastRD with $\epsilon = 0.1$. Table 2 shows that FastRD works effectively and efficiently for all the four studied deterministic networks. This again demonstrates the superiority of the algorithm FastRD in approximating the resistance diameter of huge networks with millions of nodes.

## 6.3 Performance of Algorithms for Minimizing Resistance Diameter

*6.3.1 Baseline Methods.* To evaluate the performance of our proposed algorithms, Simple, MinDiaEi, and MinDiaCH, we compared them against six baseline edge selection methods: Optimum, Random, Closeness, PageRank, Diameter, and Wspd, with details provided in the appendix.

*6.3.2 Effectiveness.* We first study the effectiveness of our algorithms Simple, MinDiaEi and MinDiaCH, by comparing them with Optimum, Random and Wspd. We execute experiments on four small real networks: Kangaroo with 17 nodes and 91 edges, Rhesus

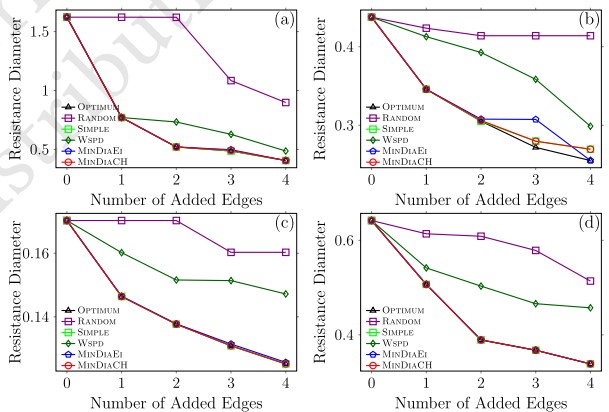

**Figure 1: Resistance diameters of the augmented graphs returned by our algorithms, as well as three baseline strategies on four networks: (a) Kangaroo, (b) Rhesus, (c) Cloister, and (d) Tribes.**

with 16 nodes and 111 edges, Cloister with 18 nodes and 189 edges and Tribes with 16 nodes and 58 edges. Their small size allows us to compute the optimal set of added edges. Since computing the optimal solution requires exponential time, we only consider $k = 0, 1, 2, 3, 4$. The results are shown in Figure 1, which shows that the resistance diameters of the augmented graphs returned by our three greedy algorithms and the optimum solutions are almost the same, all of which are much better than those returned by the Random and Wspd schemes.

In order to further demonstrate the effectiveness MinDiaEi and MinDiaCH, we also compare their returned results with five baseline schemes: Random, Diameter, Closeness, PageRank, and

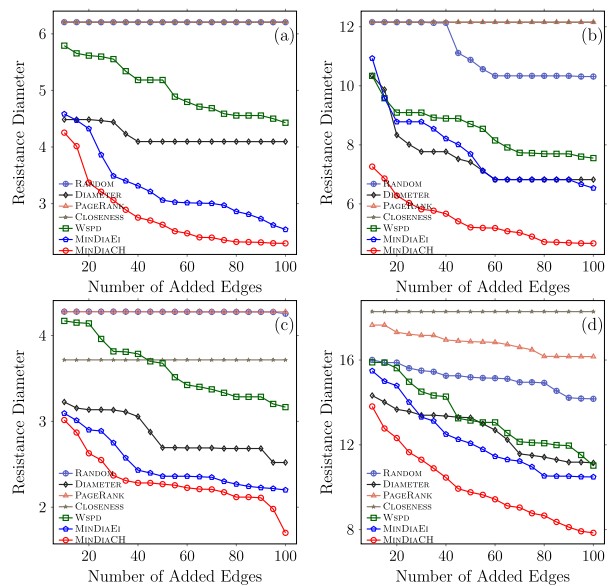

Figure 2: Resistance diameters for the augmented graphs returned by our two algorithms, and five baseline heuristics on four real netowrks: (a) Government, (b) Hep-th, (c) EmailUN, and (d) USGrid.

Wspd, on four large real networks. For each network, we calculate the resistance diameter in the original graph. Then we decrease the resistance diameter by adding up to $k = 1, 2, \ldots, 100$ new edges, applying our greedy algorithms and the five baseline strategies of edge addition. After adding each edge by different methods, we compute and record the resistance diameters. The results are shown in Figure 2, which indicates that for each network, our two greedy algorithms outperform the baseline strategies.

Finally, we execute experiments on four much larger networks to display the effectiveness of MinDiaEi and MinDiaCH. For these networks, we cannot run baselines Diameter, Closeness, PageRank, due to their high complexity, but only run Random and Wspd. The results are reported in Figure 3, which shows that MinDiaEi and MinDiaCH return significantly better results than Random and Wspd.

*6.3.3 Efficiency.* As demonstrated above, in comparison with the baseline strategies of edge addition, both of our algorithms MinDiaCH and MinDiaEi exhibit good effectiveness. Here we study the efficiency of our algorithms MinDiaCH and MinDiaEi. For this purpose, we run these two algorithms on four large-scale networks, with the largest one having over a million nodes. For each network, we select $k = 100$ and record the running time of both algorithms in Table 3. The results show that both MinDiaCH and MinDiaEi are efficient, and are scalable to networks with millions of nodes. However, there is some difference for the performance between MinDiaCH and MinDiaEi. As shown in Table 3, MinDiaEi runs faster than MinDiaCH. For example, MinDiaCH does not terminate in one day for the network Edit, while MinDiaEi outputs the solution within 2 hours. Although MinDiaEi is more efficient

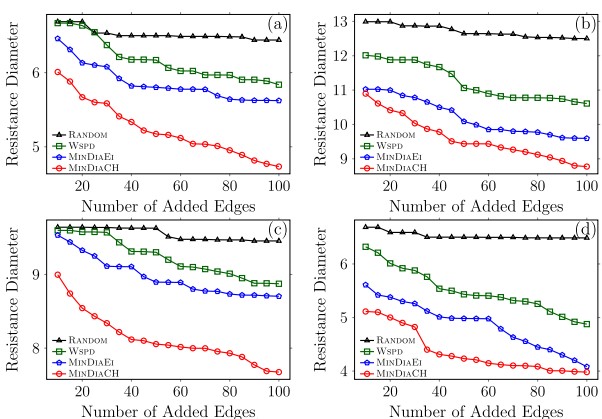

Figure 3: Resistance diameters for the augmented graphs returned by our algorithms, and two baseline schemes on four large networks: (a) HR, (b) Epinions, (c) Delicious, and (d) Edit.

than MinDiaCH, Figures 2 and 3 illustrate that MinDiaCH is more effective than MinDiaEi.

Table 3: The running time (seconds, $s$) of MinDiaCH and MinDiaEi on some large-scale networks.

| Network | Vertices | Edges | Running time ($s$) | |
|---|---|---|---|---|
| | | | MinDiaCH | MinDiaEi |
| HR | 54,573 | 498,202 | 9700 | 335 |
| Epinions | 75,877 | 508,836 | 9800 | 490 |
| Delicious | 536,108 | 1,365,961 | 71080 | 1491 |
| Edit | 1,347,094 | 5,276,371 | 168900 | 4930 |

## 7 CONCLUSION

In this paper, we present FastRD, a fast approximation algorithm for efficiently computing the resistance diameter of a graph with a guaranteed error. Direct computation of resistance diameter is computationally expensive, requiring matrix inversion and pairwise distance calculations for all nodes. To address this, FastRD employs the Johnson-Lindenstrauss Lemma, Laplacian solvers, and convex hull techniques, significantly reducing the computational complexity by focusing on estimating resistance distances for a smaller subset of node pairs. This allows the algorithm to scale effectively to large graphs with millions of nodes.

We also introduce the problem of minimizing the resistance diameter by adding $k$ edges to the graph. We proved that the problem is NP-hard and that its objective function is non-supermodular but monotone. To solve this, we propose three heuristic algorithms: Simple, MinDiaEi, and MinDiaCH, with varying trade-offs between speed and effectiveness. Extensive experiments demonstrate that all algorithms are efficient and scalable, with MinDiaEi being faster and MinDiaCH offering better results in terms of resistance diameter reduction. Future work will focus on extending these techniques to optimize other graph metrics, such as biharmonic distance.

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

## A   PROOF OF LEMMA 4.6

**Proof.** Let $u' \in C(\hat{S})$ and $v' \in C(\hat{S})$ be two points, which are the closest ones to $u$ and $v$, respectively. By Lemma 4.5 and the fact $D(S) = D(\bar{S})$, we have $d(u, u') \leq \theta D(\bar{S})$ and $d(v, v') \leq \theta D(\bar{S})$. Considering $d(x, y) = D(\hat{S})$ and the condition that $u'$ and $v'$ belong to the convex hull $C(\hat{S})$, we have $d(x, y) \geq d(u', v')$. Making use of the triangle inequality twice, it follows that $d(x, y) \geq d(u', v') \geq (d(u, v) - 2\theta D(\bar{S}))$. Note that $d(u, v) = D(\bar{S})$ and $\theta = \frac{\epsilon}{12}$, we obtain $D(\bar{S}) \geq d(x, y) \geq (1 - \frac{\epsilon}{6})D(\bar{S})$, which directly leads to (6). $\quad\square$

## B   PROOF OF LEMMA 4.7

**Proof.** Let $D(\bar{S})^2 = \max_{u,v \in \bar{S}} d(u, v)^2$. By Lemma 4.6, we have $D(\bar{S}) \geq D(\hat{S}) \geq (1 - \frac{\epsilon}{6})D(\bar{S})$. Then we obtain

$$D(\bar{S})^2 \geq D(\hat{S})^2 \geq (1 - \frac{\epsilon}{6})^2 D(\bar{S})^2$$

$$= (1 - \frac{\epsilon}{3} + \frac{\epsilon^2}{36})D(\bar{S})^2$$

$$\geq (1 - \frac{\epsilon}{3})D(\bar{S})^2$$

On the other hand, we have $\bar{R}(\mathcal{G}) = D(S)^2 = D(\bar{S})^2$. Combining the above-obtained results, we have $\bar{R}(\mathcal{G}) \geq \hat{R}(\mathcal{G}) \geq (1 - \frac{\epsilon}{3})\bar{R}(\mathcal{G})$ which leads to (7). $\quad\square$

## C   PROOF OF THEOREM 4.8

**Proof.** We first analyze the time complexity of algorithm FASTRD$(\mathcal{G}, \epsilon)$, which includes two main operations: constructing matrix $\tilde{X}$ and determining set $\hat{S}$. The construction of $\tilde{X}$ takes $\widetilde{O}(m/\epsilon^2)$ time, while finding the set $\hat{S}$ of points needs $\widetilde{O}(nl/\epsilon^2)$ time. Thus, the total running time of algorithm FASTRD is $\widetilde{O}((m + nl)/\epsilon^2)$.

We proceed to prove the correctness of the approximation error. Let $\bar{R}(\mathcal{G}) = \max_{u,v \in V} ||\tilde{X}(e_u - e_v)||_2^2$, and let $\hat{R}(\mathcal{G}) = \max_{u,v \in \hat{S}} ||\tilde{X}(e_u - e_v)||_2^2$. Then, we have $\bar{R}(\mathcal{G}) \overset{\epsilon}{\approx} R(\mathcal{G})$ and $\hat{R}(\mathcal{G}) \overset{\epsilon/3}{\approx} \bar{R}(\mathcal{G})$, both of which result in $(1 - \epsilon)R(\mathcal{G}) \leq \hat{R}(\mathcal{G}) \leq (1 + \epsilon)R(\mathcal{G})$, completing the proof. $\quad\square$

## D   HARDNESS OF RESISTANCE DIAMETER MINIMIZATION PROBLEM

The objective function of Problem 1 is not explicit, suggesting that Problem 1 seems to be difficult. In this subsection, we confirm this intuition by proving that Problem 1 is NP-hard. We will give a reduction from the 3-colorability problem, which has been shown to be NP-hard [24]. First, we introduce a minimum resistance diameter augmentation problem.

PROBLEM 2. *(Minimum Resistance Diameter Augmentation Problem) Given an undirected, connected, simple graph $\mathcal{G} = (V, E)$, a non-negative integer $k$ and a non-negative threshold $R_0$, is there a subset $P \subseteq Q$ of size $|P| \leq k$ with $Q = (V \times V) \setminus E$, such that the graph $\mathcal{H} = (V, E \cup P)$ satisfies $R(\mathcal{H}) \leq R_0$ ?*

We now show that Problem 2 is in NP. For any given graph $\mathcal{G}$ and the set $P$ of added edges, the correctness of a given solution for Problem 2 can be verified by computing the resistance distances for all $O(n^2)$ pairs of nodes, which can be obtained by inverting an associated matrix in $O(n^3)$ time. Then using (3) to get the resistance diameter and comparing the outcome with the given threshold $R_0$ verifies the solution. Therefore, the minimum resistance diameter augmentation problem is in NP.

In fact, Problem 2 is the decision version of the following optimization problem: Given an undirected, connected, simple graph $\mathcal{G} = (V, E)$ and a non-negative threshold $R_0$, find a set of currently non-existent edges of minimum size to add to $\mathcal{G}$ such that the resistance diameter of the resultant augmented graph is at most $R_0$. The main work of this section is to prove that Problem 2 is NP-hard, which immediately implies that the corresponding optimization

problem is also NP-hard. Thus, the problem of creating a fixed number of edges to a graph to minimize the resistance diameter of the augmented graph is also NP-hard.

Our proof of the NP-hardness for Problem 2 is inspired by the proof of the NP-hardness of the maximum algebraic connectivity augmentation problem [45]. For a graph $\mathcal{G} = (V, E)$ with $n$ nodes and $m$ edges, we construct a new $\mathcal{G}' = (V', E')$ as in [45]. Graph $\mathcal{G}' = (V'E')$ consists of three disjoint copies $\mathcal{G}_0, \mathcal{G}_1$ and $\mathcal{G}_2$ of graph $\mathcal{G}$. For every node $v \in V$, there is a corresponding node $v_i \in \mathcal{G}_i$ ($i = 0, 1, 2$); and for each edge $(u, v) \in E$, there is a corresponding edge $(u_i, v_i) \in \mathcal{G}_i$ ($i = 0, 1, 2$). By construction the graph $\mathcal{G}'$ has $3n$ vertices and $3m$ edges. We now consider the minimum resistance diameter augmentation problem on $\mathcal{G}'$ with $k = 3n^2 - 3m$, such that the augmented graph $\mathcal{H}$ has at most $3n^2$ edges and $R_0 = \frac{1}{n}$.

For our proof, we introduce a class of graphs. Let $\mathcal{K}_{n,n,n}$ denote the complete tripartite graph, consisting of three disjoint groups of nodes, with each having exact $n$ nodes. In $\mathcal{K}_{n,n,n}$, there is no edge linking any two nodes within the same group, while every node in one group is adjacent to every node in the other two groups. Thus, the graph $\mathcal{K}_{n,n,n}$ has $3n$ vertices and $3n^2$ edges. Using the previous result in [25], it is straightforward to obtain the resistance diameter for $\mathcal{K}_{n,n,n}$.

LEMMA D.1. *For the complete tripartite graph $\mathcal{K}_{n,n,n}$ with $3n$ nodes and $3n^2$ edges, let $R(\mathcal{K}_{n,n,n})$ be its resistance diameter. Then, $R(\mathcal{K}_{n,n,n}) = \frac{1}{n}$.*

**Proof.** In the complete tripartite graph $\mathcal{K}_{n,n,n}$, the resistance distance $r(i, j)$ between any pair of nodes $i$ and $j$ can be determined explicitly [25]. If $i$ and $j$ are in the same group, $r(i, j) = \frac{1}{n}$; If $i$ and $j$ are in different groups, $r(i, j) = \frac{3n-1}{3n^2}$. Thus, according to (3), we obtain $R(\mathcal{K}_{n,n,n}) = \max\{\frac{1}{n}, \frac{3n-1}{3n^2}\} = \frac{1}{n}$. □

In order to prove that the minimum resistance diameter augmentation problem can be reduced to the 3-colorability problem, we need the following lemmas.

LEMMA D.2. *[45] There exists a subset $P \subseteq Q$ of size $|P| \leq k$, where $Q = (V \times V) \backslash E$, such that $\mathcal{H} = (V', E' \cup P)$ is isomorphic to $\mathcal{K}_{n,n,n}$ if and only if $\mathcal{G}$ is 3-colorable.*

LEMMA D.3. *[45] For any graph $\mathcal{H} = (V, E)$ with $|V| = 3n$ nodes and $|E| \leq 3n^2$ edges for $n > 1$ satisfies $\lambda_2(\mathcal{H}) \geq 2n$ if and only if $\mathcal{H}$ is isomorphic to $\mathcal{K}_{n,n,n}$.*

LEMMA D.4. *[63] For a connected graph $\mathcal{G}$ with Laplacian matrix $L$, let $R(\mathcal{G})$ be the resistance diameter of $\mathcal{G}$, and let $\lambda_2$ be the smallest non-zero eigenvalue of $L$. Then, $R(\mathcal{G}) \leq \frac{2}{\lambda_2}$.*

LEMMA D.5. *For any graph $\mathcal{H} = (V, E)$ with $|V| = 3n$ nodes and $|E| \leq 3n^2$ edges for $n > 1$ satisfies its resistance diameter $R(\mathcal{H}) \leq \frac{1}{n}$ if and only if $\mathcal{H}$ is isomorphic to $\mathcal{K}_{n,n,n}$.*

**Proof.** Based on Lemma D.4, we have $R(\mathcal{H}) \leq \frac{2}{\lambda_2(\mathcal{H})}$. Using Lemma D.1 and Lemma D.3, if $\mathcal{H}$ is isomorphic to $\mathcal{K}_{n,n,n}$, we have $R(\mathcal{H}) = R(\mathcal{K}_{n,n,n}) = \frac{1}{n}$. On the other hand, Lemma D.3 indicates that $\lambda_2(\mathcal{H}) \geq 2n$, which, together with Lemma D.4, leads to $R(\mathcal{H}) \leq \frac{2}{\lambda_2(\mathcal{H})} \leq \frac{1}{n}$. Combining the above results, $R(\mathcal{H}) = \frac{1}{n}$ if $\mathcal{H}$ is isomorphic to $\mathcal{K}_{n,n,n}$, and $R(\mathcal{H}) > \frac{1}{n}$ otherwise. □

Lemma D.2 and Lemma D.5 directly lead to our main result of this section.

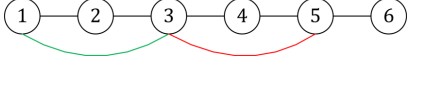

**Figure 4: A line graph with 6 nodes and 5 edges. The colored lines represent newly added edges.**

THEOREM D.6. *The minimum resistance diameter augmentation problem is NP-hard.*

# E NON-SUPERMODULAR PROPERTY OF THE OBJECTIVE FUNCTION

Two typical concepts related to an optimization problem with a cardinality constrain are monotone non-decreasing and supermodular set functions.

DEFINITION E.1. *(Monotonicity) A set function $f : 2^Q \to \mathbb{R}$ is monotone non-decreasing if $f(S) \leq f(T)$ holds for all $S \subseteq T \subseteq Q$*

DEFINITION E.2. *(Supermodularity) A set function $f : 2^Q \to \mathbb{R}$ is supermodular if*

$$f(S) - f(S \cup \{e\}) \geq f(T) - f(T \cup \{e\})$$

*holds for all $S \subseteq T \subseteq V$ and $e \in Q$.*

For a combinatorial optimization problem, when its objective function is monotone and supermodular, a simple greedy algorithm by selecting one element with the maximum marginal benefit in each iteration yields a solution with $(1 - e^{-1})$ approximation ratio [46]. However, there are still many combinatorial optimization problems whose objective functions are not supermodular. For these problems, a heuristic cannot guarantee a $(1 - e^{-1})$ approximation solution. Unfortunately, Problem 1 belongs to this problem class. Although the objective function for Problem 1 is monotone decreasing, next we show that it is non-supermodular. To this end, we give an example of the path graph $\mathcal{G}$ with 6 nodes and 5 edges in Figure 4. Let set $A = \{(1, 6)\}$, set $B = \{(1, 6), (1, 3)\}$ and edge $e = (3, 5)$. Simple computation leads to

$$R(\mathcal{G}(A)) = 1.5, \qquad R(\mathcal{G}(A \cup \{e\})) = 1.5,$$
$$R(\mathcal{G}(B)) = 1.5, \qquad R(\mathcal{G}(B \cup \{e\})) = 1.2,$$

which means

$$R(\mathcal{G}(A)) - R(\mathcal{G}(A \cup \{e\})) < R(\mathcal{G}(B)) - R(\mathcal{G}(B \cup \{e\})),$$

violating the definition of supermodularity. Therefore, the objective function of Problem 1 is non-supermodular.

# F MODEL NETWORKS AND THEIR RESISTANCE DIAMETERS

*Hierarchical Graphs.* Hierarchical graphs are constructed iteratively [50]. Starting with the complete graph $\mathcal{K}_3$ for $g = 1$, for $g \geq 2$, $\mathcal{H}(g)$ is created by generating three copies of $\mathcal{H}(g-1)$, then identifying the three vertices of a new complete graph $\mathcal{K}_3$ with the three hub nodes of each copy. Figure 5(a) illustrates the hierarchical graph $\mathcal{H}(3)$. In the hierarchical graph $\mathcal{H}(g)$, there are $3^g$ nodes

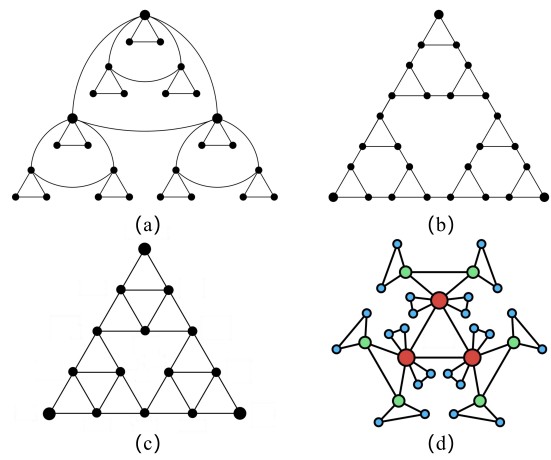

(a)                                              (b)

(c)                                              (d)

**Figure 5: Illustration of the first several iterations of four deterministic graphs: (a) the hierarchical graph $\mathcal{H}(3)$, (b) the Towers of Hanoi graph $\mathcal{T}(3)$, (c) the Sierpiński gasket $\mathcal{S}(2)$, and (d) the Koch network $\mathcal{K}(2)$.**

and $\frac{3^{g+1}-3}{2}$ edges. The resistance diameter of $\mathcal{H}(g)$ is [50]

$$R(\mathcal{H}(g)) = \frac{4(g-1)}{3} + \frac{2}{3}. \qquad (8)$$

*Hanoi Graphs.* The Hanoi graphs $\mathcal{T}(g)$ are based on the Tower of Hanoi puzzle with $g$ discs [72]. Each legal state of the puzzle corresponds to a node in $\mathcal{T}(g)$, with edges linking nodes whose states can be transformed by moving one disc; $\mathcal{T}(g)$ has $3^g$ nodes and $\frac{3(3^g-1)}{2}$ edges. Figure 5(b) illustrates the Hanoi graph $\mathcal{T}(3)$. The resistance diameter of $\mathcal{T}(g)$ has been obtained to be [54]

$$R(\mathcal{T}(g)) = \left(\frac{5}{3}\right)^g - 1. \qquad (9)$$

*Sierpiński Gaskets.* The Sierpiński gaskets are constructed iteratively. Starting with an equilateral triangle $\mathcal{S}(0)$, each iteration $g \geq 1$ involves bisecting the edges of all upward-pointing triangles in $\mathcal{S}(g-1)$ and removing the central triangle, resulting in $\mathcal{S}(g)$ with three copies of the previous iteration's triangles. Figure 5(c) illustrates the first three generations of Sierpiński gaskets. There are $\frac{3(3^g+1)}{2}$ nodes and $3^{g+1}$ edges in graph $\mathcal{S}(g)$. In [31], the resistance diameter of $\mathcal{S}(g)$ is obtained to be

$$R(\mathcal{S}(g)) = \frac{2}{3}\left(\frac{5}{3}\right)^g. \qquad (10)$$

*Koch Networks.* The Koch networks are constructed iteratively [73]. Starting with a triangle $\mathcal{K}(0)$, each iteration $g \geq 1$ involves generating two new nodes for each node in $\mathcal{K}(g-1)$, which are then connected to their "mother" nodes to form new triangles, resulting in $\mathcal{K}(g)$. Figure 5(d) illustrates the construction of the Koch network $\mathcal{K}(2)$. In network $\mathcal{K}(g)$, there are $2 \cdot 4^g + 1$ nodes and $3 \cdot 4^g$ edges. It has been shown [71] that the resistance diameter of the Koch network $\mathcal{K}(g)$ is

$$R(\mathcal{K}(g)) = \frac{2}{3}(2g+1). \qquad (11)$$

# G  OVERVIEW OF BASELINE METHODS

- OPTIMUM: choose $k$ edges from $Q$ by exhaustive search to form the set $P$ that minimize the resistance diameter of $\mathcal{G}(P) = (V, E \cup P)$.
- RANDOM: randomly choose one edge from $Q$ of the updated graph each time. Repeating this process $k$ times, until $k$ edges are added.
- CLOSENESS: choose an edge from $Q$ of the updated graph each time, the two endpoints of which have the lowest values of closeness centrality. Repeating this process $k$ times, until $k$ edges are selected.
- PAGERANK: choose an edge from $Q$ of the updated graph each time, two endpoints of which have the lowest values of PageRank centrality. Repeating this process $k$ times, until $k$ edges are selected.
- DIAMETER: choose two farthest nodes of the updated graph and establish a link between them. Repeating this process $k$ times, until $k$ edges are selected.
- WSPD: First obtain $k$ pairs of well-separated point sets in an Euclidean space generated by APPROXER, by using the well-separated pair decomposition in [12]. Let $\{\{A_1, B_1\}, \ldots, \{A_k, B_k\}\}$ be the output of WSPD. Then, choose two nodes $u_i$ and $v_i$ from each set $A_i$ and $B_i$. The $k$ edges $\{(u_1, v_1), \ldots, (u_k, v_k)\}$ are those to be added to the graph.

