# OpenReview forum: "Fast Estimation and Optimization of Resistance Diameter on Graphs"
_ACM.org/TheWebConf/2025/Conference — WWW 2025 Poster_

### Official Review · Reviewer_z4hU · 2024-11-19

**Novelty:** 6
**Technical Quality:** 6

**Review:**

This work provides methods for approximating the resistance diameter of a graph and minimizing the resistance diameter by adding a fixed number of edges. In general, the paper is well-organized and easy to read, with consistent notations and clear definitions. Theoretical analysis and extensive experiments demonstrate the efficiency and effectiveness.

**Questions:**

It is confusing for the difference between ApproxER proposed in [57] and ApproxRD in this work.

**Reviewer Confidence:**

4: The reviewer is certain that the evaluation is correct and very familiar with the relevant literature

**Scope:**

4: The work is relevant to the Web and to the track, and is of broad interest to the community

---

### Official Review · Reviewer_Xu16 · 2024-11-29

**Novelty:** 6
**Technical Quality:** 6

**Review:**

This paper proposes FastRD, a fast 𝜖-approximation algorithm to compute graph resistance diameter. It defines the resistance diameter minimization problem, which is a combinatorial problem with cardinality constraint. It then proposes two greedy heuristics to solve the problem. Experiments show the ability of FastRD to make fast estimation on large graphs.

Pros:

* FastRD is a novel algorithm that can produce scalable solutions applicable to large general graphs.
* Extensive experiments prove the effectiveness and efficiency of proposed algorithms.
* A new problem is stated and studied.
* The manuscript is overall clear and well-structured.

Cons:

* The experiments part can be improved.

**Questions:**

1. How is the error threshold set for GEER? Table 1 shows that it produces very low error rate at a high cost. Setting a higher 𝜖 can balance error & cost.
2. The running time of baseline methods for minimizing resistance diameter is not compared.

**Reviewer Confidence:**

2: The reviewer is willing to defend the evaluation, but it is likely that the reviewer did not understand parts of the paper

**Scope:**

3: The work is somewhat relevant to the Web and to the track, and is of narrow interest to a sub-community

---

### Official Review · Reviewer_Yezb · 2024-12-01

**Novelty:** 5
**Technical Quality:** 4

**Review:**

This paper explores two interconnected problems: estimating the graph resistance diameter (ERD) and reducing the resistance diameter by adding edges under a cardinality constraint (RRD).

## Estimating Graph Resistance Diameter (ERD)
The exact calculation of ERD involves two computationally intensive tasks: (1) computing the pseudoinverse of the Laplacian matrix, and (2) calculating pairwise resistance distances between \( n \) nodes. To address these challenges, the authors leverage **APPROXER**, a method introduced by Daniel A. Spielman and Nikhil Srivastava, which is based on Laplacian solvers and the JL Lemma. APPROXER approximates all pairwise resistance distances.
The authors then propose **APPROXER** and identifies the maximum among them to estimate the resistance diameter (Algorithm 2, Line 4).

Building on this foundation, the authors propose **FASTRD**, an accelerated algorithm designed to efficiently approximate the maximum resistance distance. FASTRD achieves this by focusing on approximating the farthest distances among points on the convex hull boundaries, significantly reducing the computational overhead of Line 4 in Algorithm 2.

## Reducing Resistance Diameter (RRD)
For the RRD problem, the authors establish its computational complexity, proving it to be NP-hard, monotone, but non-supermodular. To address this challenge, they present three heuristic algorithms designed to tackle the problem effectively within these constraints. Extensive experiments demonstrate the practical efficiency and effectiveness of these proposed heuristics.

## Summary
Overall, the paper provides a comprehensive approach to tackling the ERD and RRD problems, offering algorithms and validating their utility through rigorous experimentation. The paper reads very well.

**Questions:**

1. Algorithm 2 is largely derived from Lemma 4.3, with only Line 4 being original. Claiming that the authors developed Algorithm 2 might come across as overstating its novelty to the reviewers.

2. There is no introduction to the baseline GEER, which makes its inclusion somewhat confusing.

3. In the WSPD algorithm, the method for selecting \( u_i \) and \( v_i \) from \( A_i \) and \( B_i \) is not explained, leaving the baseline ambiguous.

4. The techniques and algorithms in the paper rely heavily on existing methods, which diminishes the technical novelty of the work.

5. As the graph size increases, the relative error in estimating RD also grows. For instance, on the HR, Epinions, and Slashdot datasets, the relative errors are as high as 0.87. While the authors demonstrate low relative errors on four model networks, the performance on real datasets is less impressive and raises concerns about practical reliability.

**Reviewer Confidence:**

3: The reviewer is confident but not certain that the evaluation is correct

**Scope:**

4: The work is relevant to the Web and to the track, and is of broad interest to the community

---

### Official Review · Reviewer_oiUg · 2024-12-02

**Novelty:** 5
**Technical Quality:** 5

**Review:**

A new approximate algorithm is proposed for the calculation problem of "resistance diameter" and the optimization problem of graphs, and its superiority is proved through experiments, which is highly innovative.

Insufficient background details: Although it is mentioned that "resistance diameter" has a wide range of applications, no detailed description of specific application scenarios or practical impacts is given. For some readers, more background information may be needed to understand the importance of this problem.

**Questions:**

1. Please introduce the symbols in the introduction in detail
2. Please introduce the necessity of this study

**Reviewer Confidence:**

2: The reviewer is willing to defend the evaluation, but it is likely that the reviewer did not understand parts of the paper

**Scope:**

3: The work is somewhat relevant to the Web and to the track, and is of narrow interest to a sub-community

---

### Official Review · Reviewer_gKm7 · 2024-12-03

**Novelty:** 5
**Technical Quality:** 6

**Review:**

Pros.
1. The paper introduces a novel approach to approximating the resistance diameter, which is a significant contribution to the field, especially considering the computational complexity of direct computation.
2. The authors provide a solid theoretical foundation for their algorithms, including error bounds and complexity analysis, which adds credibility to their claims.
3. The paper demonstrates the practical applicability of the algorithms through extensive experiments on real-world and model networks, showcasing scalability and effectiveness.

Cons.
1. The paper focuses on resistance diameter, and it's unclear how these methods could generalize to other graph metrics or problems.
2. The paper does not address how the algorithms perform on dynamic or streaming graphs, which is an important consideration for many real-world applications.

**Questions:**

See Cons.

**Reviewer Confidence:**

3: The reviewer is confident but not certain that the evaluation is correct

**Scope:**

4: The work is relevant to the Web and to the track, and is of broad interest to the community